

# An autoregressive integrated moving average and long short-term memory (ARIM-LSTM) hybrid model for multi-source epidemic data prediction

Benfeng Wang[1], Yuqi Shen[1], Xiaoran Yan[2] and Xiangjie Kong[1]

[1] College of Computer Science and Technology, Zhejiang University of Technology, Hangzhou, Zhejiang, China
[2] The Research Institute of Artificial Intelligence, Zhejiang Lab, Hangzhou, Zhejiang, China

## ABSTRACT

The COVID-19 pandemic has far-reaching impacts on the global economy and public health. To prevent the recurrence of pandemic outbreaks, the development of short-term prediction models is of paramount importance. We propose an ARIMA-LSTM (autoregressive integrated moving average and long short-term memory) model for predicting future cases and utilize multi-source data to enhance prediction performance. Firstly, we employ the ARIMA-LSTM model to forecast the developmental trends of multi-source data separately. Subsequently, we introduce a Bayes-Attention mechanism to integrate the prediction outcomes from auxiliary data sources into the case data. Finally, experiments are conducted based on real datasets. The results demonstrate a close correlation between predicted and actual case numbers, with superior prediction performance of this model compared to baseline and other state-of-the-art methods.

## INTRODUCTION

On July 9, 2020, the World Health Organization (WHO) proclaimed the COVID-19 outbreak a pandemic (*Shahid, Zameer & Muneeb, 2020*). According to *World Health Organization (WHO) (2023)* on May 4, 2023, COVID-19 no longer constituted an international public health emergency. Nowadays, COVID-19 has been classified as a Class B infectious disease, and management methods akin to Class A infectious diseases are being implemented (*Li et al., 2020*). The three-year pandemic caused severe damage to the global economy and people's lives and properties. These devastations underscored that society would do well to invest now in creating prevention models for the next pandemic (*Caulkins et al., 2023*).

The epidemic development trend prediction has become a central focus for global scholars, involving traditional mathematical and deep learning models. *Benvenuto et al. (2020)* employed an autoregressive integrated moving average (ARIMA) model to forecast the epidemiological trends. Meanwhile, *Iqbal et al. (2021)* employed a long short-term

Corresponding author
Xiangjie Kong, xjkong@ieee.org

memory (LSTM) model to predict the percentage of positive patients in Pakistan for June 2020. *Kong (2021)* utilized the ARIMA model to interpolate the incidence data of dengue fever in Zhejiang Province over the past 15 years and established the relationship between dengue fever incidence and meteorological factors using the LSTM model. *Li, Chen & Yang (2022)* proposed a novel hybrid forecasting model named GVMD-ELM-ARIMA, based on Gradient Variational Mode Decomposition (GVMD), extreme learning machine (ELM), and autoregressive integrated moving average (ARIMA), to enhance the prediction accuracy of cumulative COVID-19 confirmed cases. These works demonstrate that hybrid models exhibit superior performance compared to singular traditional mathematical or deep learning prediction models. Traditional mathematical models excel in handling linear prediction tasks with long-term dependencies, such as seasonality. In contrast, deep learning models exhibit significant advantages in capturing more complex nonlinear patterns, particularly in tasks involving short-term fluctuations. The transmission process of pandemics presents characteristics of periodic dependencies and dynamically complex variations, making it challenging for a single prediction model to forecast pandemic trends accurately. Thus, integrating traditional mathematical models with deep learning holds paramount research significance for pandemic trend prediction tasks.

The data sources for epidemic forecasting tasks are diverse, encompassing case data, sentiment text data, and travel mobility data. *Tiwari, Kumar & Guleria (2020)* utilized confirmed case data in India to predict the number of confirmed cases, cured cases, and deaths. They predicted when the epidemic would peak and made prevention recommendations. *Chang & Nguyen (2022)* employed flight review data from TripAdvisor to forecast the impact of COVID-19 on the aviation industry. *Schlosser et al. (2020)* used mobile phone mobility data to predict changes in mobile network structure resulting from lockdowns. The centralization of mobile networks caused the peak of the epidemic outbreak in advance, but it helped reduce the epidemic peak. The privacy concerns surrounding epidemic case data and its scarcity challenge prediction task accuracy. While auxiliary data sources such as sentiment text data and mobility data contain vast amounts of information, they may not directly reflect the dynamics of epidemic spread. Thus, leveraging auxiliary data sources to support case data and undertake prediction tasks on epidemic trends, enhancing prediction accuracy, has emerged as a new challenge.

Therefore, we aim to predict future epidemic trends in a multi-source data scenario. To address this issue, we propose a multi-source data fusion-based ARIMA-LSTM model. First, we forecast future trends based on case data and population mobility data using the ARIMA-LSTM model. Subsequently, we integrate the prediction results of auxiliary data sources into the case data through a fully connected layer based on the Bayes-Attention mechanism. Finally, we experiment with real datasets to validate the model's effectiveness. Our main contributions are threefold:

- We propose a pandemic future case trend prediction model based on ARIMA and LSTM, which combines the advantages of both linear and nonlinear models to enhance prediction performance.

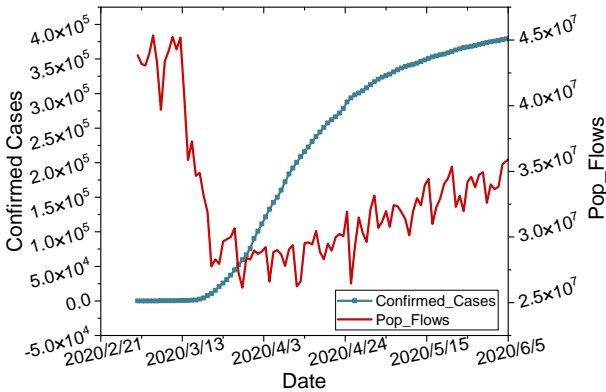

**Figure 1** Time series changes in confirmed cases and population flow data.

- We design a fully connected layer based on a Bayes-Attention mechanism to fusion multi-source data, thereby enhancing the accuracy of predictions.
- We conduct a series of experiments on real epidemic datasets, including hyperparameter evaluation, model comparison, and ablation studies. The results indicate that our prediction model achieves high accuracy, with predicted confirmed cases closely matching the actual confirmed cases.

The rest of this article is as follows: In 'Related Work', we provide the related work for this work. 'Proposed Method' describes a hybrid method involving ARIMA-LSTM model and Bayes-Attention machanism. 'Experiments' analyzes the model's training process and experimental results. 'Conclusion' concludes the article.

## RELATED WORK

Epidemics are one of the most serious issues in public health, characterized by their suddenness and seasonality in transmission (*Merow & Urban, 2020*). The suddenness leads to nonlinear growth in infection cases. The seasonality causes linear periodic changes in transmission peaks. Hybrid models combining ARIMA and LSTM can handle such linear and nonlinear tasks. Furthermore, the pandemic has prompted governments to implement lockdown measures, resulting in changes in population mobility. Simultaneously, intense population movements due to factors such as holidays have stimulated peaks in the pandemic. Diverse datasets exhibit a notable correlation, as shown in Fig. 1. Multi-source data sets can complement each other through such correlations. Therefore, we will introduce related work from hybrid models and multi-source data.

### Hybrid models

Researchers have applied various techniques to forecast the epidemic pattern, involving time series models and deep learning models. Time series models such as ARIMA excel in handling linear and periodic tasks. *Mahalle et al. (2020)* presented medical insights into COVID-19 and used the Prophet model for forecasting. Due to its open-source

algorithms, accuracy, and faster data fitting, they recommend Prophet for forecasting. Subsequently, *Kumar & Susan (2020)* forecasted epidemics in ten countries using ARIMA and Prophet time series prediction models. They demonstrated that the ARIMA model in predicting morbidity rates is more effective. *Kumar & Viral (2021)* discussed the impact of COVID-19 on the Indian economy, society, and financial sectors using the ARMA model. They illustrated the future scenarios of India under the influence of epidemic in the coming months. *Alabdulrazzaq et al. (2021)* forecasted the confirmed and recovered cases in various stages of Kuwait's gradual prevention plan using the ARIMA model. Such works all have predicted the future trends of the epidemic. However, the accuracy of the epidemic forecasts is not high. It is unsuitable in complex and dynamic environments.

Deep learning models, such as LSTM, excel at pattern recognition and prediction. *Oliveira, Gruetzmacher & Teixeira (2021)* utilized artificial neural networks (ANN) to forecast the confirmed cases and death counts for the next 7 days in Brazil, Portugal, and the United States time series. *Chandra, Jain & Singh Chauhan (2022)* employed recurrent neural networks (RNN) for multi-step short-term infection forecasting. *Chimmula & Zhang (2020)* used the LSTM model to predict the trends of the epidemic and the likely cessation time. They suggested COVID-19 might cease around June 2020. *Kong et al. (2024)* proposed a multi-modal traffic forecasting framework, MPGNNFormer, to predict human mobility patterns, addressing issues in both long-term and short-term scenarios. Such works achieved predictions for specific case numbers of the epidemic. However, they are suitable for handling short-term volatile tasks and do not have an advantage for long-term dependent tasks.

Hence, researchers have begun exploring the application of hybrid time series and deep learning models in epidemic prediction. *Jin et al. (2022)* proposed a parallelized approach based on a regression coefficient-weighted ARIMA-LSTM model to forecast epidemic data in China. They concluded that the epidemic in China would exhibit a steady downward trend over the next 60 days. *Jin et al. (2023)* modeled and forecasted the pandemic in Quebec, Canada, using a hybrid LSTM and ARIMA model. Through comparisons of three combined forecasting models, they demonstrated that CNN-ARIMA-LSTM exhibited the best prediction performance. These methods have been proven to have higher accuracy than single LSTM or ARIMA prediction models. Furthermore, we are preparing to enhance the prediction accuracy of the ARIMA-LSTM model through the integration of multidimensional data fusion.

## Multi-source data

The data sources for pandemic prediction tasks are often diverse, encompassing case data, sentiment text data, and travel mobility data. *Wang et al. (2020)* utilized case data to perform time series predictions of the pandemic trends in Russia, Peru, and Iran. *Chakraborty & Ghosh (2020)* developed a novel hybrid autoregressive integrated moving average-wavelet-based forecasting (ARIMA-WBF) model using case data from multiple countries, including Canada, France, India, South Korea, and the UK, to predict and assess risks. While case data can directly forecast the development trends of epidemics, the limited quantity of data poses constraints on the accuracy of predictions. In addition, *Alamoodi et*

*al. (2021)* conducted impact analysis and opinion mining on epidemic and other infectious diseases through sentiment text data on social media. *Dubey (2020)* performed national-level sentiment analysis based on Twitter comments during the pandemic. Sentiment text data is commonly employed for analyzing the societal impact of pandemics, and its reflection on the epidemic exhibits a lag effect. *Alessandretti (2022)* estimated real-world large-scale contact networks and developed scenario predictions to understand the driving factors of disease spread through mobile GPS data. *Gan et al. (2020)* assessed risks during the pandemic by collecting mobile traffic data. *Zhan et al. (2020)* modeled and predicted epidemic based on population migration data. These data cannot directly reflect the development of the pandemic. Yet, they can serve as supplementary for assessment and prediction. According to *Kong et al. (2022)*, there is a significant correlation between the pandemic and population mobility. Therefore, we will design a method that integrates case data and population mobility data to enhance the accuracy of our prediction model.

## PROPOSED METHOD

This section will provide a detailed description of the proposed methodology. Our approach involves utilizing an ARIMA-LSTM model designed for predicting multi-source epidemic data. Firstly, we employ the ARIMA-LSTM model to forecast the development trends of epidemic case data and population flow data. Secondly, we introduced a fully connected layer based on the Bayes-Attention mechanism to integrate predictions from multi-source data. Finally, we utilize various evaluation metrics to assess the prediction performance. Specifically, we will elaborate on the following four aspects: the ARIMA-LSTM model, the FC layer, and evaluation indicators. The architecture of the proposed model is shown in Fig. 2.

### ARIMA-LSTM model

The ARIMA-LSTM model is used to predict the epidemic development trend. To address the long-term dependency and dynamic complexity issues in epidemic spread, we consider integrating traditional mathematical models and deep learning models to enhance prediction accuracy. Firstly, we segment the time series into low-volatility and high-volatility sequences using a moving average method. Secondly, we stabilize the sequence through differencing integration operations (I) and predict future values using an autoregressive moving average (ARMA) model for the low-volatility time series. Thirdly, we forecast future values using a long short-term memory (LSTM) model for the high-volatility time series. Finally, we add the prediction results of high-volatility and low-volatility sequences, resulting in the final sequence of case predictions.

**Data partition:** The moving average method can segment time series into low-volatility and high-volatility sequences involving simple moving average (SMA), exponential moving average (EMA), and weighted moving average (WMA). SMA excels in handling sequences with consistent weight coefficients. WMA is proficient in dealing with sequences where weight coefficients vary linearly with time intervals. EMA is adept at managing sequences where weight coefficients exhibit exponential changes over time intervals. We will assess the suitability of the current moving average method based on error values, thereby selecting

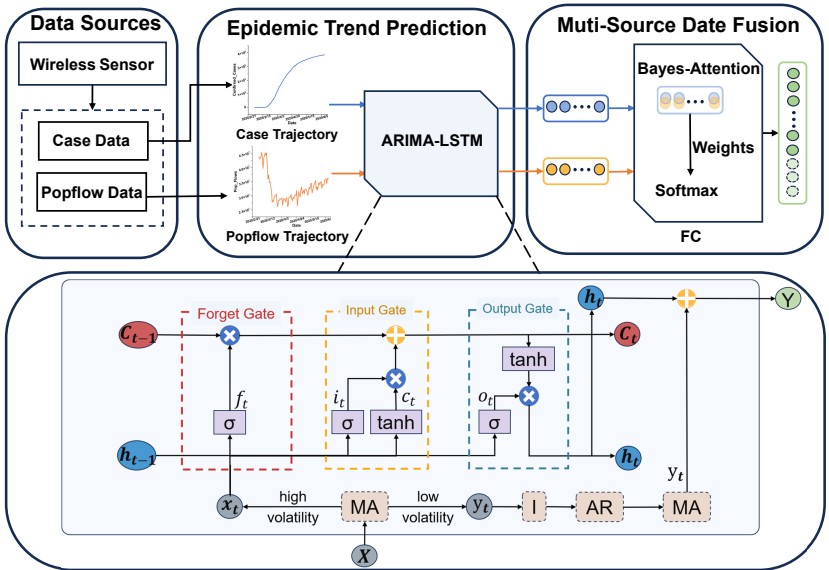

**Figure 2** **The architecture of the multi-source ARIMA-LSTM model.**

the optimal moving average method. The calculation of the moving average method is as follows:

$$SMA(t) = \frac{1}{N} \sum_{i=1}^{N} P_{t-i}, \tag{1}$$

$$WMA(t) = \frac{\sum_{i=1}^{N} w_i \cdot P_{i-i}}{\sum_{i=1}^{N} w_i}, \tag{2}$$

$$EMA(t) = \alpha \cdot P_t + (1-\alpha) \cdot EMA(t-1). \tag{3}$$

In the above equations, $N$ denotes the size of the moving average window, and $P_{t-i}$ represents the value at time $t-i$. $w_i$ signifies the weight coefficient, $\alpha$ denotes the smoothing factor, typically within the range of $(0,1)$.

**ARIMA model:** Autoregressive integrated moving average model, commonly known as ARIMA$(p, d, q)$, involves parameters $p$, $d$, and $q$, representing the order of the autoregressive (AR) terms, the order of differencing (I) required to achieve stationarity in the time series, and the order of the moving average (MA) terms, respectively. The modeling and forecasting process of the ARIMA model is illustrated in Fig. 3. The ARIMA model forecasts future values by linearly calculating past observed values and time error terms of time series data. Therefore, the ARIMA model is particularly advantageous for data with clear linear trends and periodicity, making it suitable for predicting seasonal cyclic variations in epidemics.

Differential Integration (I) operation can transform a non-stationary sequence into a stationary one, typically involving first-order or higher-order differences. The order of

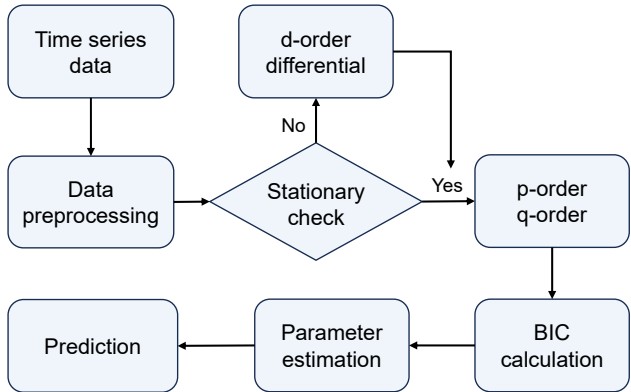

**Figure 3** **The modeling and forecasting process of the ARIMA model.**

differencing integration $d$ can be determined through visualization methods by selecting the order that minimizes the fluctuations in the time series.

Autoregression (AR) term expresses the relationship between current and historical values. It is used to predict the current value based on its past values. The formula for a p-order autoregressive process is defined as follows:

$$y_t = \mu + \sum_{i=1}^{p} \gamma_i y_{t-i} + \varepsilon_t, \tag{4}$$

where $y_t$ is the current value, $\mu$ is a constant term, $p$ is the order, $\theta_i$ represents the autoregressive coefficients, and $\varepsilon_t$ is the error term.

Moving average (MA) term represents the relationship between the current value and random errors, focusing on the accumulation of error terms in the autoregressive model. It effectively eliminates random fluctuations in predictions. The formula for a q-order moving average process is calculated as

$$y_t = \mu + \sum_{i=1}^{q} \theta_i \varepsilon_{t-i} + \varepsilon_t, \tag{5}$$

where $y_t$ is the current value, $\mu$ is a constant term, $q$ is the order, $\theta_i$ represents the moving average coefficients, and $\varepsilon$ is the error term.

Therefore, we can express the computational formula for ARIMA$(p,d,q)$ as follows:

$$y_t = \mu + \sum_{i=1}^{p} \gamma_i y_{t-i} + \sum_{i=1}^{q} \theta_i \varepsilon_{t-i} + \varepsilon_t. \tag{6}$$

To determine the optimal values of $p$ and $q$, we computed the Bayesian Information Criterion (BIC). The BIC calculates the probability function and adds a penalty term for the number of parameters, which helps to avoid overfitting and provides a balanced approach to model selection. The calculation formula is described as

$$BIC = k\ln(n) - 2\ln(L), \tag{7}$$

where $k$ is the number of parameters, $n$ is the sample size, $L$ is the likelihood function. A smaller BIC value indicates that the given parameters provide a more accurate model description.

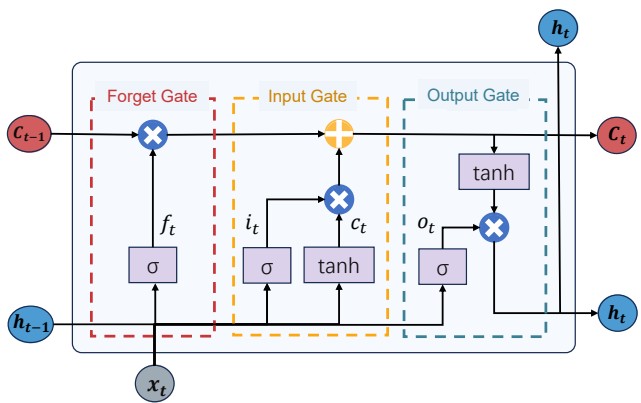

**Figure 4 The structure of the LSTM model.**

**LSTM model:** The long short-term memory (LSTM) model is a distinctive recurrent neural network (RNN). It addresses issues such as gradient vanishing and exploding commonly encountered in traditional recurrent neural networks while providing long-term and short-term memory capabilities. The LSTM model integrates a state structure and three gate structures: the cell state, forget gate, input gate, and output gate, facilitating the dynamic adjustment of self-recurrent weights. The structure of the LSTM model is depicted in Fig. 4. This structure enables the LSTM model to dynamically learn and adjust its internal parameters to adapt to different time series data patterns and features. The LSTM model can retain relevant information over extended periods, making it suitable for predicting complex and evolving tasks during epidemic outbreaks. Next, we will proceed to offer an in-depth introduction to the gated structure of the LSTM model:

The forget gate determines which information from the previous time step's cell state should be forgotten or discarded. It utilizes a sigmoid activation function to produce an output between 0 and 1, where 0 signifies complete forgetting, and 1 indicates complete retention. This aids the network in forgetting irrelevant information, mitigating gradient exploding and vanishing issues. The calculation formula for the forget gate $F_t$ is calculated as:

$$f_t = \sigma \left( W_f \cdot \left[ h_{t-1}, x_t \right] + b_f \right), \tag{8}$$

where $\sigma$ is the sigmoid activation function, $W_f$ is the weight matrix for the forget gate, $h_{t-1}$ is the previous time step's hidden state, $x_t$ is the input at the current time step, and $b_f$ is the bias term for the forget gate.

The input gate regulates the extent to which new input data flows in. It incorporates a sigmoid activation function and its output ranges between 0 and 1. The role of the input gate is to determine which information should be updated and added to the cell state. If the output of the input gate is close to 1, the network considers this information necessary and should be incorporated into the cell state. The calculation process for the input gate is as follows:

$$c_t = tanh \left( W_c \cdot \left[ h_{t-1}, x_t \right] + b_c \right), \tag{9}$$

$$i_t = \sigma \left( W_i \cdot [h_{t-1}, x_t] + b_i \right), \tag{10}$$

$$C_t = f_t * C_{t-1} + i_t * c_t. \tag{11}$$

In the above equation, $c_t$ is the candidate cell state, $W_c$ is the weight matrix for the cell state, and $b_c$ is the bias term for the cell state. $i_t$ is the output of the input gate, $W_i$ is the weight matrix for the input gate, and $b_i$ is the bias term for the input gate.

The output gate regulates the output from the cell to the hidden state. It employs a sigmoid activation function to determine which parts of the cell state should be output. Simultaneously, it uses a tanh activation function to ensure that the output values $h_t$ are within the range of $-1$ to 1. The role of the output gate is to generate the final output of the LSTM unit based on the current information in the cell state. The calculation process is as follows:

$$o_t = \sigma \left( W_o \cdot [h_{t-1}, x_t] + b_o \right), \tag{12}$$

$$h_t = o_t * tanh(C_t). \tag{13}$$

In the above equation, $o_t$ is the output gate's output, $W_o$ is the weight matrix for the output gate, $b_o$ is the bias term, and $h_t$ is the final output.

Finally, we add the prediction results of the high-volatility series and the low-volatility series to get the final case prediction series.

$$Y = y_t + h_t, \tag{14}$$

where $x_t$ represents the forecast result of the low-volatility ARIMA model, and $h_t$ represents the forecast result of the high-volatility LSTM model.

## FC layer

In the fully connected layer, we introduce a Bayes-Attention fusion mechanism. This mechanism can dynamically allocate attention weights among multiple data sources and integrate auxiliary data sources into the case data. Specifically, we first dynamically allocate attention weights among various data sources through the attention mechanism. Subsequently, based on these attention weights, we integrate auxiliary data sources into the case data through a Bayesian network. Let $Y = \{Y_1, Y_2, \ldots, Y_T\}$ be the connection vector generated from multi-source data. The calculation formula for the Bayesian attention fusion mechanism is as follows:

$$M = \tanh(Y), \tag{15}$$

$$\alpha = softmax(w^T \cdot M), \tag{16}$$

$$P_i = \frac{\alpha_i \cdot P(Y_i|Y)}{\sum_{i=1}^{T} \alpha_i \cdot P(Y_i|Y)}, \tag{17}$$

$$p = Y \cdot P^T. \tag{18}$$

In the above equation, $w$ represents the learned parameter vector, which maps the hidden states to attention weights. $\alpha$ denotes the attention weights, and $P$ signifies the conditional probability distribution.

Finally, to facilitate comparisons with other baseline and state-of-the-art models in the experimental section, We process the fused dataset through a *softmax* function to concentrate its values within the range $[0, 1]$.

$$y = softmax(p). \tag{19}$$

## Evaluation indicators

To compare the prediction effects of different models more intuitively, we use the root-mean-squared error (RMSE), the mean absolute percentage error (MAPE), the mean absolute error (MAE), and the r-squared score ($R^2$). RMSE, MAPE, and MAE are commonly used evaluation metrics based on the errors between actual and predicted values. $R^2$ is used to assess the goodness of fit of a model to the data. As our model not only needs to fit complex epidemic transmission scenarios but also maintain prediction accuracy, we will evaluate the model's goodness of fit to multi-source data using $R^2$ and assess the prediction accuracy using RMSE, MAPE, and MAE.

**RMSE:** Root-mean-squared error (RMSE) is the square root of the mean of the squared differences between predicted and actual values. It demonstrates a remarkable sensitivity to prediction errors, exerting a more substantial penalty on significant mistakes than minor discrepancies. The more minor the RMSE, the smaller the gap between predicted and actual values, indicating a more accurate model. The formula for calculating RMSE is as follows:

$$RMSE = \sqrt{\frac{1}{n}\sum_{i=1}^{n}\left(\hat{y}_i - y_i\right)^2}. \tag{20}$$

**MAPE:** Mean absolute percentage error (MAPE) represents the average percentage error of each observed value. It expresses errors in percentage form, making it more easily understandable. The smaller the MAPE, the smaller the prediction error, indicating higher model accuracy. The calculation formula for MAPE is as follows:

$$MAPE = \frac{100\%}{n}\sum_{i=1}^{n}\left|\frac{\hat{y}_i - y_i}{y_i}\right|. \tag{21}$$

**MAE:** Mean absolute error (MAE) is the average of the absolute differences between predicted and actual values. Unlike RMSE, MAE does not consider the square of errors,

**Table 1  Specific details of the dataset.**

| The cumulative confirmed case dataset | | | | | | | |
| --- | --- | --- | --- | --- | --- | --- | --- |
| ProvinceState | 4/25/21 | 4/26/21 | 4/27/21 | 4/28/21 | 4/29/21 | 4/30/21 | 05/01/21 |
| Alabama | 526131 | 526348 | 526707 | 527083 | 527513 | 527922 | 528472 |
| California | 3794549 | 3796285 | 3798103 | 3801766 | 3799797 | 3804036 | 3806052 |
| NewYork | 2031093 | 2034102 | 2037414 | 2044345 | 2040448 | 2048150 | 2052307 |
| Washington | 395312 | 397417 | 398509 | 401718 | 400149 | 403040 | 404586 |
| The daily population mobility dataset | | | | | | | |
| $geoid_o$ | $geoid_d$ | $lng_o$ | $lat_o$ | $lng_d$ | $lat_d$ | date_range | pop_flows |
| 1001 | 1001 | −86.64 | 32.53 | −86.64 | 32.53 | 03/09/20 | 86486 |
| 1001 | 1015 | −86.64 | 32.53 | −85.83 | 33.77 | 03/15/20 | 562 |
| 1001 | 1051 | −86.64 | 32.53 | −86.15 | 32.60 | 04/05/20 | 147182 |
| 1001 | 1007 | −86.64 | 32.53 | −87.13 | 33.00 | 05/17/20 | 84 |

making its treatment of large and small errors relatively equal. The smaller the MAE, the smaller the absolute difference between the actual and predicted values, indicating a more accurate model prediction. The formula for calculating MAE is as follows:

$$MAE = \frac{1}{n}\sum_{i=1}^{n}\left|\hat{y}_i - y_i\right|. \tag{22}$$

**$R^2$**: R-squared ($R^2$) represents the extent to which the model explains the variability of the target variable. It evaluates how well a model fits the data. It ranges from 0 to 1, with values closer to 1 indicating a better fit of the model. An $R^2$ of 1 signifies a perfect fit to the data, while an $R^2$ of 0 implies that the model cannot explain the variability of the target variable. The calculation formulas for $R^2$ is as follows:

$$R^2 = 1 - \frac{\sum_{i=1}^{n}\left(\hat{y}_i - y_i\right)^2}{\sum_{i=1}^{n}\left(\hat{y}_i - \bar{y}\right)^2}. \tag{23}$$

In the above formulas, n is the total sample size, $\hat{y}_i$ is the predicted value of the model, $y_i$ is the true value, $\bar{y}$ is the average of the true value.

## EXPERIMENTS

### Datasets

The experiments of our proposed model were completed in Python 3.11.6. This study utilizes the cumulative confirmed case numbers in New York State from March 2, 2020, to February 24, 2023, and daily population mobility trajectory numbers from February 15, 2020, to October 15, 2022, for modeling, prediction, and analysis. The data were sourced from the Johns Hopkins University website and the U.S. COVID-19 pandemic population mobility dataset, both publicly available. The datasets are shown in Table 1.

Upon acquiring the data, the outliers were cleaned through time series diagrams, and mean substitution treatment was applied for outlier handling. We utilized the first 80% of the data as the training set for the model and the remaining 20% as the test set to assess the model's generalization ability.

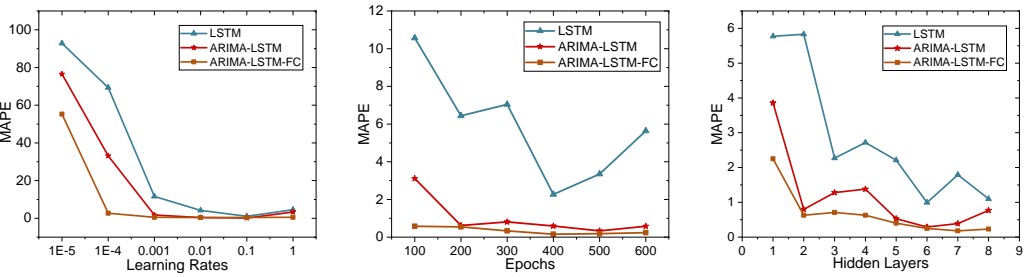

**Figure 5** Time series diagram comparing the predicted and true values of the ARIMA-LSTM-FC model.

**Table 2** ARIMA-LSTM-FC model parameters.

| Parameter | Value |
|---|---|
| (p, d, q) | (1, 2, 1) |
| Hidden layers | 7 |
| layer_num | 2 |
| Epochs | 400 |
| Learning rate | 0.01 |
| Optimizer | Adam |

## ARIMA-LSTM-FC model

This study initially trains the ARIMA-LSTM model using cumulative confirmed case data from New York, NY, USA. The hyperparameter training process for the model is depicted in Fig. 5. Due to MAPE representing the mean percentage error and exhibiting superior performance, we utilize MAPE to assess the training effectiveness of the model. Subsequently, future trend predictions are derived separately based on confirmed case data and population mobility data. Finally, the prediction outcomes of population mobility data are integrated into the case data through a fully connected (FC) layer utilizing the Bayes-Attention mechanism.

The parameters for the ARIMA-LSTM-FC model are configured as specified in Table 2. The time series graph comparing the predicted values is shown in Fig. 6.

## Comparison experiment

The ARIMA-LSTM-FC hybrid model exhibits better prediction accuracy based on the model prediction graphs. We introduce several baseline and advanced models to provide a more precise comparison of the prediction performance. These time series forecasting models include LSTM, Bi-LSTM, GRU, Transformer, and CNN-ARIMA-LSTM.

**LSTM:** *Shahid, Zameer & Muneeb (2020)* utilized LSTM, GRU, and Bi-LSTM models to forecast the time series of confirmed cases, deaths, and recoveries in 10 major countries. We implement the model methodologies from their work and perform prediction tasks based on our dataset.

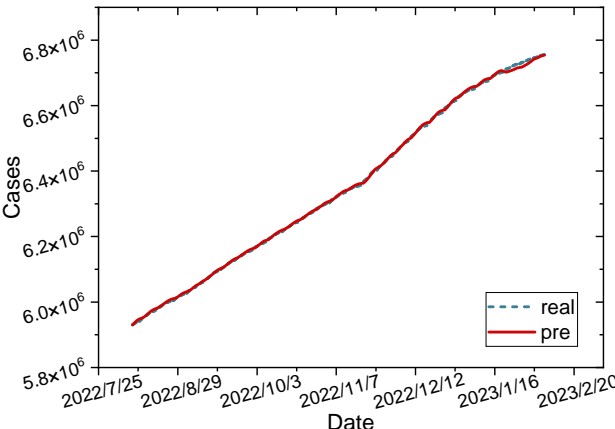

**Figure 6** Time series diagram comparing the predicted and true values of ARIMA-LSTM-FC model.

**Bi-LSTM:** The bidirectional long short-term memory (*Shahid, Zameer & Muneeb, 2020*) (Bi-LSTM) is an extension of the LSTM network designed for processing sequential data. It introduces two independent LSTM layers at each time step: one for forward propagation and the other for backward propagation. Unlike the standard LSTM network, Bi-LSTM can consider both past and future information of sequence data simultaneously, enabling a more comprehensive understanding and modeling of the context of sequential data.

**GRU:** The gated recurrent unit (GRU) (*Shahid, Zameer & Muneeb, 2020*) is a variant of the RNN designed for processing sequential data. It controls the flow and update of information in sequence data through gate mechanisms, including update and reset gates, thereby modeling long-term dependencies. The GRU model has fewer parameters, faster training speed, and better generalization capabilities.

**Transformer:** The Transformer is a deep learning model based on attention mechanisms, with its core ideas including the self-attention mechanism and positional encoding. Its modular structure and parallel computation capabilities address the efficiency issues of RNNs when dealing with long sequences.

**CNN-ARIMA-LSTM:** *Jin et al. (2023)* proposed three combined forecasting models, namely CNN-LSTM-ARIMA, TCN-LSTM-ARIMA, and SSA-LSTM-ARIMA, to predict the pandemic in Italy. Ultimately, they demonstrated that the CNN-LSTM-ARIMA model performed the best. Therefore, we conduct comparative experiments using the CNN-LSTM-ARIMA model.

We evaluate the performance of each model using evaluation metrics such as the root-mean-squared error (RMSE), the mean absolute percentage error (MAPE), the mean absolute error (MAE), and the determination coefficient ($R^2$). The prediction performance evaluation metrics of each model on the test set are shown in Table 3.

Based on the four evaluation metrics selected in this study, we compare the performance of various models. We denote the best-performing metric in bold and the second-best metric underlined. Among them, the ARIMA-LSTM-FC model demonstrates the best prediction performance, followed by the CNN-ARIMA-LSTM model. Comparative

**Table 3** Model evaluation metrics.

| Model | RMSE | MAPE | MAE | $R^2$ |
|---|---|---|---|---|
| LSTM | 0.18 | 8.49 | 0.076 | 0.089 |
| Bi-LSTM | 0.158 | 5.92 | 0.053 | 0.524 |
| GRU | 0.16 | 4.56 | 0.05 | 0.86 |
| Transformer | 0.14 | 15.21 | 0.14 | 0.544 |
| CNN-ARIMA-LSTM | 0.063 | 1.75 | 0.054 | 0.91 |
| ARIMA-LSTM-FC | **0.022** | **0.42** | **0.032** | **0.993** |

**Notes.**
   Values in bold indicate the best-performing metric. An underline indicates the second-best metric.

experimental results indicate that our proposed multi-source data fusion-based ARIMA-LSTM model can better address complex epidemic transmission scenarios, further enhancing prediction accuracy.

## Ablation study

To validate that the hybrid model performs better than individual models, we conducted ablation experiments through the following three parts: the ARIMA model, the LSTM model, and the ARIMA-LSTM model. As our fully connected layer is intended to incorporate the prediction outcomes from auxiliary data sources into the case data, the resulting trend still reflects the future trajectory of the case data. We will conduct ablation experiments based on case data to ensure a consistent dimension for comparing experimental results.

**ARIMA model:** We prioritize the model training using the cumulative confirmed cases. Initially, the original data undergoes differencing. After applying a second-order difference, the signal sequence becomes stationary and white noise-free. The parameters (p, d, q) of the model and the white noise nature of the residual were determined using the Bayesian information criterion (BIC). A smaller BIC value indicates a more precise model description based on the parameters. The calculation process is illustrated in Eq. (7). The results of the BIC are depicted in a heatmap, as shown in Fig. 7.

The red mark on the heatmap shows that the final model is ARIMA (2,2,2). Finally, the comparison between the ARIMA model's predicted values and the actual values in the time series is depicted in Fig. 8.

**LSTM model:** The LSTM model is constructed using the cumulative confirmed cases in New York State from March 2, 2020, to February 24, 2023. Both the input and output of the model are cumulative confirmed case numbers. To determine the parameters of the LSTM model, we conducted hyperparameter experiments, as shown in Fig. 5.

Finally, the model parameters are configured as specified in Table 4.

Based on the specified parameters, we construct the LSTM model. The time series graph comparing the predicted values of the LSTM model with the actual values is shown in Fig. 9.

**ARIMA-LSTM model:** We use the cumulative confirmed cases as a variable to construct the ARIMA-LSTM model. We retrain the model and visualize the training process of

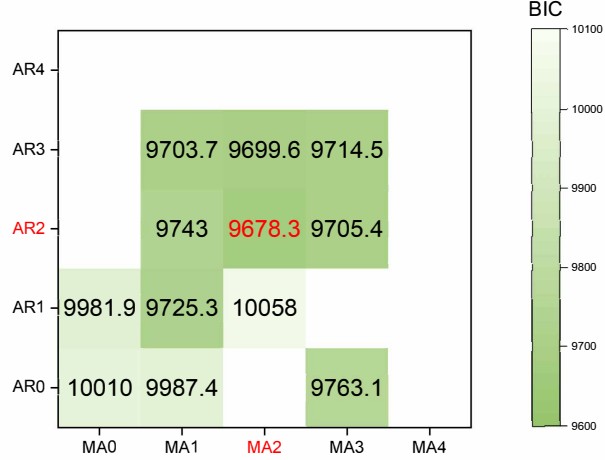

**Figure 7** The heatmap of BIC.

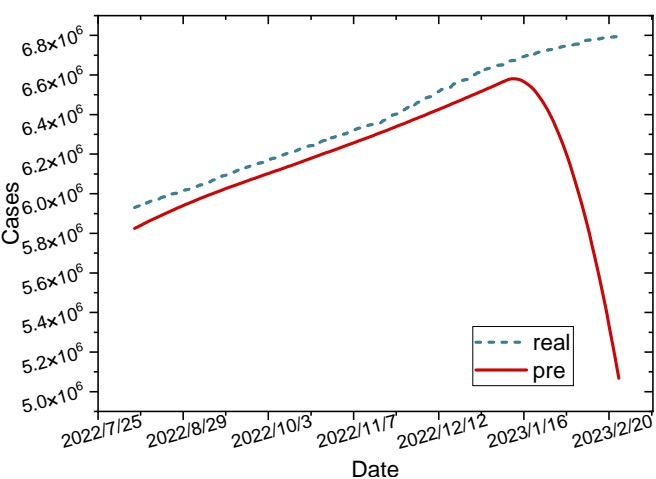

**Figure 8** Time series diagram comparing the predicted and true values of the ARIMA model.

**Table 4** LSTM model parameters.

| Parameter | Value |
| --- | --- |
| Hidden layers | 6 |
| layer_num | 2 |
| Epochs | 400 |
| Learning rate | 0.1 |
| Optimizer | Adam |

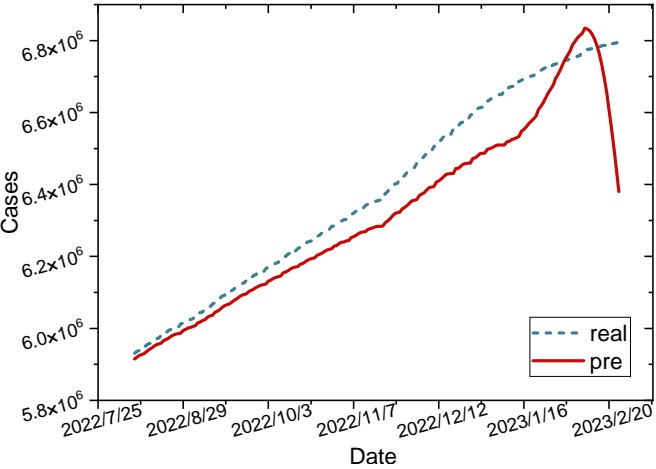
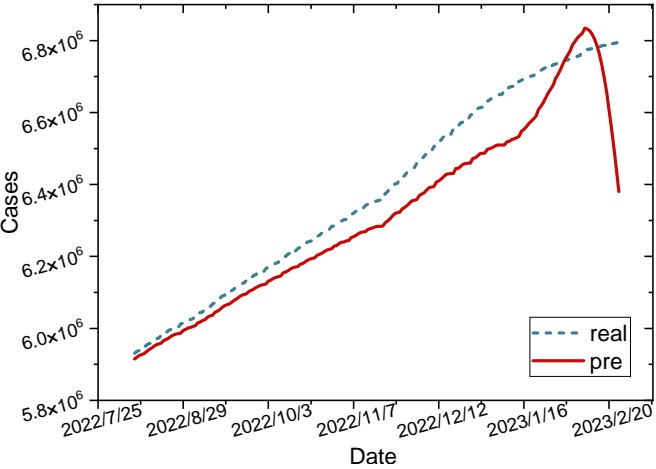

**Figure 9** Time series diagram comparing the predicted and true values of the LSTM model.

**Table 5 ARIMA-LSTM model parameters.**

| Parameter | Value |
| --- | --- |
| (p, d, q) | (1, 2, 1) |
| Hidden layers | 6 |
| layer_num | 2 |
| Epochs | 500 |
| Learning rate | 0.1 |
| Optimizer | Adam |

hyperparameters in Fig. 5. The parameters for the hybrid model are configured as specified in Table 5.

Once the parameters are configured, we train the ARIMA-LSTM model using the training set and then utilize the trained model to make predictions on the test set. The time series graph comparing the predicted values of the ARIMA-LSTM model with the actual values is illustrated in Fig. 10.

According to the comparative analysis of prediction results from various models, it is evident that the ARIMA-LSTM-FC model indeed exhibits superior forecasting accuracy. We still use evaluation metrics such as the root mean square error (RMSE), the mean absolute percentage error (MAPE), the mean absolute error (MAE), and the determination coefficient ($R^2$) to make a more precise comparison. The performance evaluation metrics for each model on the test set are presented in Table 6.

Based on four evaluation criteria, we compare the performance of our proposed models. Here, we denote the best-performing metric with bold and the second-best metric with underline. Among them, the ARIMA-LSTM-FC model demonstrates the best prediction performance, with ARIMA-LSTM following closely. Since the ARIMA-LSTM-FC model integrates multiple data sources, its performance surpasses that of the ARIMA-LSTM model. Moreover, due to the combination of linear and nonlinear methods, the prediction

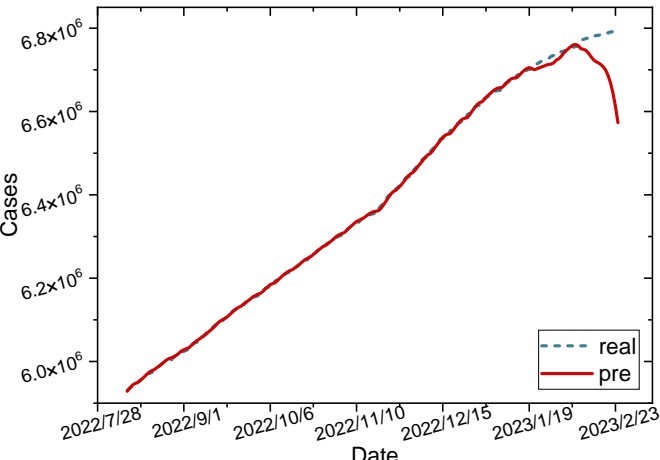

**Figure 10  Time series diagram comparing the predicted and true values of ARIMA-LSTM model.**

**Table 6  Ablation experiment results.**

| Model | RMSE | MAPE | MAE | R² |
|---|---|---|---|---|
| ARIMA | 0.066 | 4.83 | 0.044 | 0.396 |
| LSTM | 0.18 | 8.49 | 0.076 | 0.089 |
| ARIMA-LSTM | 0.058 | 2.33 | 0.038 | 0.88 |
| ARIMA-LSTM-FC | **0.022** | **0.42** | **0.032** | **0.993** |

**Notes.**
Values in bold indicate the best-performing metric. An underline indicates the second-best metric.

performance of the ARIMA-LSTM model exceeds that of standalone LSTM or ARIMA models. Through the fully connected layer, we integrate the prediction outcomes from auxiliary data sources into the case data. This approach ensures that both the input and output of the model consist of case data, which is crucial for the effectiveness of ablation experiments. The results of ablation experiments indicate that the hybrid prediction model and multi-source data fusion indeed enhance prediction performance.

## CONCLUSION

The transmission of epidemics exhibits periodic dependency and dynamic complexity characteristics, posing significant challenges to forecasting efforts. We propose an ARIMA-LSTM model based on multi-source data fusion to predict the epidemic's development trends more accurately. Initially, we employ the ARIMA-LSTM model to forecast trends in multiple data sources separately. Subsequently, we introduce a fully connected (FC) layer based on a Bayesian attention mechanism to integrate the prediction outcomes from auxiliary data sources into the case data. The computational metrics for the ARIMA-LSTM-FC model are as follows: RMSE = 0.022, MAPE = 0.42, MAE = 0.032, $R^2$ = 0.993. Among them, all these metrics outperform those of other compared models. Experimental results demonstrate that the prediction performance of the ARIMA-LSTM-FC model

surpasses that of the ARIMA-LSTM model, and the ARIMA-LSTM model's prediction performance exceeds that of the standalone LSTM or ARIMA models. Thus, the fusion of the ARIMA-LSTM model with multi-source data enhances prediction performance.

Our data fusion method, anchored in Bayesian networks and attention mechanisms, dynamically allocates attention weights for integrating multiple data sources without imposing any constraints on the data. The basic principle of the ARIMA-LSTM hybrid model lies in predicting future values based on historical data. Numerous studies have already explored the application of the ARIMA-LSTM model across various domains, demonstrating its versatility across different scenarios. Thus, the multi-source data fusion-based ARIMA-LSTM model we propose exhibits universal applicability.

The COVID-19 pandemic has surpassed the period covered in this article. However, the outbreak of the pandemic has undoubtedly sounded an alarm worldwide, emphasizing the need to invest in prediction tasks related to epidemic development trends. The multi-source data-fused ARIMA-LSTM forecasting model established in this article can serve as a reference for governments of various countries engaged in similar infectious disease control efforts, aiming to minimize the losses caused by pandemics. In the future, we have plans to propose more efficient models for the fusion of multi-source data and explore predicting the development trends of epidemics in countries where case data is lacking.

### Funding

This work was supported by the National Natural Science Foundation of China (No.62072409), the National Key Research and Development Program of China (No.2022YFF0712400), the Zhejiang Provincial Natural Science Foundation (No.LR21F020003), and the Fundamental Research Funds for the Provincial Universities of Zhejiang (No.RF-B2020001). There was no additional external funding received for this study. The funders had no role in study design, data collection and analysis, decision to publish, or preparation of the manuscript.

### Grant Disclosures

The following grant information was disclosed by the authors:
The National Natural Science Foundation of China: No. 62072409.
The National Key Research and Development Program of China: No. 2022YFF0712400.
The Zhejiang Provincial Natural Science Foundation: No. LR21F020003.
The Fundamental Research Funds for the Provincial Universities of Zhejiang: No. RF-B2020001.

### Competing Interests

Xiangjie Kong is an Academic Editor for PeerJ. The other authors declare that they have no competing interests.

## Author Contributions

- Benfeng Wang conceived and designed the experiments, performed the experiments, analyzed the data, performed the computation work, prepared figures and/or tables, authored or reviewed drafts of the article, and approved the final draft.
- Yuqi Shen conceived and designed the experiments, performed the experiments, performed the computation work, prepared figures and/or tables, authored or reviewed drafts of the article, and approved the final draft.
- Xiaoran Yan conceived and designed the experiments, performed the experiments, analyzed the data, performed the computation work, prepared figures and/or tables, authored or reviewed drafts of the article, and approved the final draft.
- Xiangjie Kong conceived and designed the experiments, performed the experiments, analyzed the data, performed the computation work, prepared figures and/or tables, authored or reviewed drafts of the article, and approved the final draft.

## Data Availability

The code and data are available at Github and Zenodo:

- https://github.com/Bevan-Wang/MEHP

- Bevan_Wang. (2024). Bevan-Wang/MEHP: Code for epidemic trend prediction (AL-Forecast). Zenodo. https://doi.org/10.5281/zenodo.10980889.

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
