# Peer review of "An autoregressive integrated moving average and long short-term memory (ARIM-LSTM) hybrid model for multi-source epidemic data prediction"

_PeerJ Computer Science, doi:10.7717/peerj-cs.2046_

## Round 0.1 · original submission · Major Revisions

Please follow the reviewer's recommendations, especially reviewer 1.

Reviewer 1 ·

Basic reporting

The paper is well written, understandable and uses a good English style. Raw data (code) is provided and complete.
I believe there are some issues that need to be resolved:
1. The introduction correctly introduces the objectives of the paper, but the background does not reflect quite a few of the contributions combining LSTM and ARIMA. The rest of the references are significant and appropriated.
2. Figures are correct and relevant, but a more detailed description would be needed in the text of the paper related to figures 3 and 6.

Experimental design

Some aspects should be improved:
1. The paper presents an interesting idea, although not a completely novel one. The multi-source prediction model is novel but not the hybridization ARIMA-LSTM.
2. The idea of using multi-source data is very interesting, but its use in the proposed model should be better explained and its contribution should be explained in more detail.
3. The Fully Connected (FC) layer of the model should be better described.
4. The evaluation indicators used in the model should be justified, or at least commented.
5. I wonder if there is not some error in the ablation study, since it talks about both the ARIMA-LSTM model and the ARIMA-LSTM-FC model. It is clear that until you apply the FC layer you do not get the final unified results, so there really is no ARIMA-LSTM component result. This should be adequately clarified in the paper.

Validity of the findings

Conclusions are consistent with the results and the planed objectives, beyond the misunderstanding regarding ARIMA-LSTM and ARIMA-LSTM-FC.

Reviewer 2 ·

Basic reporting

The abstract needs to be improved, with a focus on introducing the content of this study.

Experimental design

1. The main content of the article is to use FC to combine the ARIMA-LSTM model. So, how should the ARIMA-LSTM model mentioned in Figure 4 be combined?
2. Simply optimizing the ARIMA-LSTM model with FC lacks innovation and further improvement is needed. Lack of innovation, significant modifications are needed here.

Validity of the findings

The conclusion should provide a more detailed explanation of which indicators of the methods used in the article are better than other models, and whether this result is universal.

·

Basic reporting

Authors proposed an ARIMA-LSTM (Autoregressive Integrated Moving Average and Long Short-Term Memory ) model for predicting future cases and utilize multi-source data to enhance prediction performance.

Experimental design

Please see additional comments.

Validity of the findings

Please see additional comments.

Additional comments

Reviewer Comments:
1. List of keywords missing
2. Proposed work flow diagram missing
3. Dataset details can be specified in a table
4. What is the training-testing ratio?

Above are my comments.
Paper can be accepted with revisions.

---

## Round 0.2 · accepted · Accept

Congratulations. Two reviewers agreed the paper is ready for publication.

Reviewer 2 ·

Basic reporting

no comment

Experimental design

no comment

Validity of the findings

no comment

Additional comments

The author has made modifications according to the premise's suggestions. I suggest accepting the article.

·

Basic reporting

clear

Experimental design

well defined

Validity of the findings

Results provided

Additional comments

author made changes based on review comments